# Survival analysis of early initiation of antenatal care visits and associated factors among pregnant women in Ethiopia

**Nuru Mohammed Hussen**[1]*, **Kindu Kebede Gebre**[2], **Tilahun Gemechu Rorisa**[1], **Mekash Ayalew Mohammed**[3]

1 Department of Statistics, Samara University, Semera, Ethiopia, 2 Department of Statistics, College of Computing and Informatics, Haramaya University, Dire Dawa, Ethiopia, 3 Department of Mathematics, Samara University, Semera, Ethiopia

* nurediin5111@gmail.com

## Abstract

### Introduction

Early initiation of antenatal care visits creates an opportunity for early detection of high-risk pregnancies, pregnancy complications, health education, and counseling about successful care and nutrition of the mother and the fetus. Thus, the main objective of this study was to identify the factors associated with the early initiation of antenatal care visits in Ethiopia.

### Methods

The study was conducted based on the children's data set of the 2019 Ethiopia mini-demographic and health survey. The study employed a retrospective cohort study design using the data downloaded from the Measure Demographic and Health Survey website. The study included a random sample of 2922 pregnant women who had complete information about their first antenatal care visits. A gamma-accelerated failure time model was employed to identify the factors associated with the early initiation of antenatal care visits in Ethiopia.

### Results

In Ethiopia, 62% of the pregnant women started their first antenatal care visit early. The higher birth order of the current pregnancy ($\phi$ = 2.215, 95% CI: 1.8901, 2.5966), mothers being rural residents ($\phi$ = 1.239, 95% CI: 1.1633 1.3195), mothers aged 25–34 ($\phi$ = 1.692, 95% CI: 1.5588, 1.8369), and mothers aged above 34 ($\phi$ = 1.826, 95% CI: 1.6392, 2.0336) were associated with an accelerated time to initiation of antenatal care visits. However, mothers attained higher education ($\phi$ = 0.830, 95% CI: 0.7345, 0.9373) and rich wealth index ($\phi$ = 0.869, 95% CI: 0.8156, 0.9259) were associated with a shorter time to initiation of antenatal care visits.

### Conclusion

A higher proportion of urban and educated pregnant women initiated their antenatal care visits earlier than the other groups. We recommend that stakeholders at the federal and

**Data Availability Statement:** The datasets relevant to this paper are publicly available from the Measure DHS website at https://www.dhsprogram.

com/data/dataset/Ethiopia_Interim-DHS_2019.
cfm?flag=1.

**Funding:** The author(s) received no specific
funding for this work.

**Competing interests:** The authors have declared
that no competing interests exist.

**Abbreviations:** AFT, Accelerated Failure Time; AIC,
Akaike Information Criteria; ANC, Antenatal Care;
BIC, Bayesian Information Criteria; CSA, Central
Statistical Agency; EA, Enumeration Area; EMDHS,
Ethiopia Mini Demographic and Health Survey;
EPHI, Ethiopian Public Health Institute; FMoH,
Federal Ministry of Health; ICF, International
Classification of Functioning; PH, Proportional
Hazard; SNNPR, Southern Nations, Nationalities,
and Peoples' Region; WHO, World Health
Organization.

regional levels should focus on providing special concern for information, education, and
communication regarding the importance of early antenatal care visits initiation.

## Introduction

Antenatal care (ANC) visits are a vital component of maternal healthcare that aid in the identification of high-risk pregnancies, the prevention and treatment of any complications, emergency preparedness, birth planning, meeting any unmet nutritional, social, emotional, and physical needs of pregnant women, and providing patient education, including successful care and nutrition of the newborn [1]. Even though substantial progress has been reported over the past two decades, the global burden of pregnancy-related causes and stillbirths is still high [2]. Globally, more than 303000 women died from preventable pregnancy and pregnancy complications; 2.7 million babies died during the first 28 days of life; and 2.6 million babies were stillborn [3]. However, Sub-Saharan Africa and Southern Asia accounted for the highest burden (86%) of global maternal deaths [4]. Recent evidence indicates that a higher frequency of antenatal care visits with pregnant women is associated with a reduced likelihood of stillbirths and maternal deaths due to complications [3].

The World Health Organization's (WHO) recommendation for starting an antenatal care visit is in the first trimester, around or preferably before week 12 of pregnancy [1]. This stage of pregnancy is the fastest developmental period of the fetus, in which all organs become well developed and need special care [5]. Moreover, the revised WHO report recommended that each pregnant mother have a minimum of eight antenatal visits, where the first visit should take place in the first trimester (up to 12 weeks of gestation), two visits should take place in the second trimester (at 20 and 26 weeks of gestation), and five visits should take place in the third trimester (at 30, 34, 36, 38, and 40 weeks) [1,6]. Increasing the number of antenatal care visits to eight or more can reduce prenatal deaths by up to 8 per 1000 births when compared to 4 visits [3]. However, only 64% of pregnant women obtain antenatal care visits four or more times [3].

The WHO recommended four or more visits among pregnant women in low- and middle-income countries, where the first visit should be between week 8 and 12, the second visit should be between weeks 24 and 26, the third visit should be at week 32, and the fourth visit should be between 36 and 38 weeks of pregnancy and return for delivery at week 41 if the women doesn't give birth [6]. Moreover, an evidence-based report of the World Health Organization (WHO) recommended eight rather than four antenatal care visits among pregnant women in low- and middle-income countries [6].

In Ethiopia, the burden of maternal death due to pregnancy and its complications was 412 maternal deaths per 100,000 live births [7]. Even though antenatal care visits are free of charge, late initiation of antenatal care visits is still a problem [8]. The Ethiopia Ministry of Health recommendation for early initiation of antenatal care visits follows the trend of the World Health Organization recommendation for low- and middle-income countries [7]. Evidence indicated that only 28% of pregnant women had their first antenatal care visit within the first 3 months, while 43% of women in urban areas received antenatal care visits within the first three months of pregnancy, compared with 22% of pregnant women in rural areas [9]. Moreover, a study conducted in northwest Ethiopia showed that 52% of pregnant women attended their first antenatal care visit after 16 weeks of pregnancy [10]. The government of Ethiopia, together with the ministry of health, initiates activities like enhanced coordination of health extension

workers, the health development army, and extended supportive supervision systems at national, regional, zonal, Woredas, primary healthcare units, and community levels on maternal health services to improve early initiation of antenatal care visits [11]. The studies conducted on the early initiation of antenatal care visits showed that it is significantly constrained due to a number of factors like the age of the mother, level of income, place of residence, unwanted pregnancy, level of education, and distance to a health facility [10,12,13]. However, the factors associated with early initiation of antenatal care visits may vary across the different socio-economic and cultural statuses of mothers, and the use of an appropriate statistical methodology is also an important issue. In Ethiopia, a number of quantitative studies were done on the factors of early and late initiation of antenatal care visits using a binary logistic regression model [11,14–17], but in addition to dichotomizing this variable as early and late initiation, it is important to include the exact first time the mother goes to health facilities to get care from health professionals. Thus, in view of the above literature, the main objective of this study was the identification of the factors associated with early initiation of antenatal care visits through the appropriate survival analysis procedure, using the data from the 2019 Ethiopia Mini Demographic and Health Survey (EMDHS).

## Materials and methods

### Data source and study design

This paper uses secondary data from the children's data set of the 2019 Ethiopia Mini Demographic and Health Survey (EMDHS). The study employed a retrospective cohort study design using the data obtained from the Measure DHS Website (https://www.dhsprogram.com/data/dataset/Ethiopia_Interim-DHS_2019.cfm?flag=1) after reasonable request, and data use permission was obtained from the institutional review board of the International Classification of Functioning (ICF). The survey was carried out by the Ethiopian Public Health Institute (EPHI) in partnership with the Central Statistical Agency (CSA) and the Federal Ministry of Health (FMoH), with technical assistance from the ICF and financial and technical support from development partners, and conducted from March 21 to June 28, 2019.

### Sampling design

The sampling frame for the 2019 Ethiopia mini demographic and health survey was a complete list of the 149,093 enumeration areas (EAs), which was performed by the Central Statistical Agency (CSA). The average coverage for an enumeration area was 131 households. The survey used a two-stage stratified sampling design. Ethiopia is structured administratively into nine nation-states and two administrative cities, which are then stratified into urban and rural except for Addis Ababa, which is entirely urban, to establish 21 sampling strata. To ensure the precision of sample selection, the allocation was done through an equal allocation where 25 EAs were selected from eight regions and each of the three main regions was assigned 35 EAs: Amhara, Oromia, and the Southern Nations, Nationalities, and Peoples' Region (SNNPR). Among the 305 EAs selected in the primary stage, 93 were urban and 212 were rural, and in the second stage of selection, an equal probability systematic selection was used to select a fixed number of 30 families per cluster from the newly established household listing. 8,885 women of reproductive age (15–49) were eligible and interviewed from a nationally representative sample of 9012 women. However, this study excluded the women that didn't provide an information about their first time of antenatal care visits, so it consists of a weighted sample of 2922 women.

## Variables in the study

**Response variable.**　The response variable in this study was the timing of antenatal care visits, measured in months. Early initiation of antenatal care visits was considered as the event of the study, and the response time is based on the most recent recommendations for early initiation of antenatal care visits among pregnant women in low-and middle-income countries [18]. Censoring was considered when pregnant women started their first antenatal visit after the first three months of pregnancy.

**Explanatory variables.**　The selection of explanatory variables was theoretically driven and supported by prior research with regard to factors influencing the early initiation of antenatal care visits in Ethiopia. Accordingly, age of mothers, mother's education level, marital status, wealth index, birth order of a child, preceding birth interval, and place of residence were considered as the potential explanatory variables of early initiation of antenatal care visits.

## Data processing and analysis

The data was managed with SPSS 26 and analyzed using SAS 9.4. The data was weighted to take into account or adjust for disproportionate sampling and non-sampling responses and to increase the representativeness of the sample. Variable selection was made by the purposeful method, while a univariable analysis was made between a response variable and each predictor separately at a 25% level, then a multivariable analysis was performed at a 5% level between a response variable and all significant variables in the univariable analysis. The model selection was performed using Akaike Information Criteria (AIC) and Bayesian Information Criteria (BIC) due to the presence of non-nested models.

## Ethics approval and consent to participate

This study was a secondary data analysis of the Ethiopia Mini Demographic and Health Survey (EMDHS). The institutional review board (IRB) of the International Classification of Functioning (ICF) approved the procedures for DHS public-use datasets that don't in any way allow respondents, households, or sample communities to be identified. It approved the study entitled "**Early initiation of antenatal care visits and associated factors in Ethiopia.**" The authors confirmed that all the procedures were followed in accordance with relevant guidelines and regulations. The DHS report confirmed that ethical consent was obtained from all eligible subjects (women aged 15–49 and men aged 15–59); the remaining data about children was obtained from their respective legal guardians.

## Survival analysis

Survival analysis is the analysis of statistical data when the response variable is the time until an event occurs.

**Cox proportional hazards model.**　The most commonly used model relating time to event outcome with the number of explanatory variables is the proportional hazards (PH) model, proposed by Cox (1972) [19]. The Cox proportional hazard model is based on the assumption of proportional hazards (the hazard ratio is constant over survival time), and no specific probability distribution is assumed for the survival times. The model relates the hazard function h(t,x,β) as a function of the baseline hazard function $h_0(t)$ and the vector of explanatory variables r(x,β). Then, the Cox proportional hazards model is expressed as;

$$h(t, x, \beta) = h_0(t)\exp(x^t\beta) \quad \text{Where,}$$

h(t,x,β) = The hazard function at time t, $h_0(t)$ = the baseline hazard function that characterizes

how the hazard function changes as a function of survival time; explanatory variables x = {$x_1$, $x_2, x_3, \ldots \ldots, x_n$}, and exp($x^t\beta$), characterizes how the hazard function changes as a function of a sets of explanatory variables.

The basic assumption in survival analysis is the proportional hazards assumption. It states that the hazard ratios are constant over the survival times. If this assumption is satisfied the common survival analysis like Cox PH model are more likely to be employed. But if this assumption is violated parametric accelerated failure time models will be fitted. The global test is one of the commonly used methods to test the proportional hazards assumption. The hypothesis under the global test states that; $H_0$: the proportional hazard assumption is not violated versus $H_1$: the proportional hazard assumption is violated.

**Parametric survival models.** Parametric survival models are a class of models where a specific probability distribution is assumed for the survival times. The most commonly used probability distributions assumed for survival times are the exponential distribution, Weibull distribution, gamma distribution, log-normal distribution, and log-logistic distributions [20].

*Exponential distribution.* This distribution is also called a memoryless distribution, where the hazard function is assumed to be constant over time. Let T be the survival time that follows an exponential distribution with parameter λ.then the probability density function of T is given by:

$$f(t) = \lambda e^{-\lambda t}, \; where \; t \geq 0, \; \lambda \geq 0$$

Then the survivor function is S (t) = $e^{-\lambda t}$ and the hazard function is given by h (t) = λ

*The Weibull distribution.* It is parametrized as both the proportional hazards model and the accelerated failure time model. This distribution has a hazard function that is monotonically increasing, decreasing, or constant with time [21]. The probability density function for the Weibull distribution with the scale (λ) and shape (γ) parameters is given by:

$$f(t) = \lambda \gamma t^{\gamma-1} e^{-\lambda t^\gamma} \quad \lambda, \; \gamma > 0$$

Then the survivor function is S (t) = $e^{-\lambda t^\gamma}$ and the hazard function is given by h (t) = $\lambda\gamma t^{\gamma-1}$
Where and are the scale and shape parameters.

*Log-logistic distribution.* The log-logistic distribution is preferable when there is a non-monotonic hazard function; rather, the hazard function may increase at early times and decrease at later times. The probability density function for the log-logistic distribution is defined as:

$$f(t) = \frac{e^\theta k t^{k-1}}{(1 + e^\theta t^k)^2} \quad for \; 0 \leq t < \infty, \; k > 0$$

Then the survivor function is S (t) = $(1+e^\theta t^k)^{-1}$ and the hazard function is given by

$$h(t) = \frac{e^\theta k t^{k-1}}{1 + e^\theta t^k}$$

*Log-normal distribution.* This distribution is appropriate for random variables with positive values. A random variable T is said to have a lognormal distribution with parameters μ and σ, if log (T) has a normal distribution with μ and a standard deviation $\delta$.

The probability density function of T is given by;

$$f(t) = \frac{1}{\delta\sqrt{2\pi}} t^{-1} e^{-\frac{1}{2}\frac{(-\log(t)-\mu)^2}{\delta^2}} \quad where \; 0 \leq t < \infty, \; \delta > 0$$

Then the survivor function is then S (t) = 1- $\Phi\left(\frac{\log t-\mu}{\sigma}\right)$ and the hazard function is given by h

(t) = $\frac{\phi\left(\frac{\log t}{\sigma}\right)}{1-\Phi\left(\frac{\log t}{\sigma}\right)\sigma t}$ where $\Phi$(t) is the cumulative density function.

*The gamma distribution.* The probability density function of a gamma distribution with mean $\rho/_\lambda$ and variance $\rho/_{\lambda^2}$ is such that:

$$f(t) = \frac{\lambda^\rho t^{\rho-1}e^{-\lambda t}}{\Gamma(\rho)} \text{ for } 0 \leq t < \infty, \quad \lambda > 0, \text{ and } \rho > 0$$

The survival function is then S(t) = 1-$\Gamma_{\gamma t}(\rho)$ and the hazard function is given by h(t) = $^f(t)/_s(t)$, where $\Gamma_{\gamma t}(\rho)$ is known as the incomplete gamma function, given by;

$$\Gamma_{\lambda t}(\rho) = \int_0^{\lambda t} u^{\rho-1} e^{-u} du$$

**Accelerated failure time models.** Accelerated failure time (AFT) models are alternative models for the analysis of survival data in which explanatory variables measured on an individual are assumed to act multiplicatively on the timescale, and they measure the direct effect of each explanatory variable on the survival times [22]. These models are less affected by the choice of probability distributions than the Cox proportional hazards model. However, the most frequently used probability distributions in accelerated failure time models are the exponential distribution, Weibull distribution, log logistic distribution, log normal distribution, and gamma distribution. Suppose there is the values of p explanatory variables recorded for each individual in a study, then the hazard function of the $i^{th}$ individual is,

$$h_i(t) = h_0(t/e^{-\eta_i})e^{-\eta_i}$$

Where $\eta_i = \alpha_1 x_{1i} + \alpha_2 x_{2i} + \alpha_3 x_{3i} + \ldots\ldots\ldots + \alpha_p x_{pi}$

*Gamma accelerated failure time model.* Suppose that survival times are assumed to have a gamma distribution with the shape parameter k and the scale parameter θ, then the hazard function is given by;

$$h_T(t) = \frac{\theta}{(1 + (k-1)\int_0^\infty s^{k-2}e^{-\theta t(s-1)}ds)}$$

This function is increasing if the shape parameter (k) >1. Decreasing if k<1, and constant if k = 1. The gamma distribution can be reduced to exponential distribution as the survival time is large. Then from the above hazard function the gamma AFT model have the following probability density function;

$$f_{T|X}\left(\frac{t}{x}\right) = \frac{\theta^k}{\Gamma(k)}t^{(k-1)}e^{-\theta t}, \quad \text{Where } \theta = \frac{1}{exp(\alpha_0 + \alpha'x)}$$

## Results

### Socio-demographic and economic characteristics of mothers

Among the random sample of 2922 pregnant women, 62% have got their first antenatal care visit in the first 3 months of pregnancy, and the remaining 38% got their checkup late, from the fourth month to the date of delivery. The median survival time for the first antenatal check

**Table 1. Socio-demographic and economic characteristics of mothers.**

| Variable | Categories | Frequency (%) | Early ANC initiation (%) |
|---|---|---|---|
| Initiation of ANC | Early | 1812(62.0) | 1812(100) |
| | Delayed | 1109(38.0) | 0(0) |
| Residence | Urban | 767(26.2) | 571(74.5) |
| | Rural | 2155(73.8) | 1241(57.6) |
| Age | 15–24 | 755(25.8) | 494(65.5) |
| | 25–34 | 1558(53.3) | 976(62.6) |
| | >34 | 609(20.8) | 342(56.2) |
| Education | No education | 1298(44.4) | 679(52.3) |
| | Primary | 1147(39.3) | 753(65.6) |
| | Secondary | 318(10.9) | 238(74.8) |
| | Higher | 158(5.4) | 143(90.5) |
| Marital status | single | 29(1.0) | 22(75.9) |
| | married | 2720(93.1) | 1690(62.1) |
| | Other | 173(5.9) | 100(57.8) |
| Wealth index | poor | 890(30.4) | 433(48.7) |
| | middle | 644(22.1) | 386(59.9) |
| | rich | 1388(47.5) | 993(71.5) |

Key: ANC, Antenatal Care visit.

was 4 months, which indicates that the probability that these mothers will not get their first antenatal visit for three months or more is 0.5. Among these mothers with early initiation of antenatal care visits, the majority were urban residents (571, or 74.5%), with the age group 15–24 years (494, or 65.5%), attained higher education (143, or 90.5), being single (22, or 75.9%), and having the rich wealth index (993, or 71.5%). The modal values for birth order and preceding birth interval for the current pregnancy were second order and 34 months, respectively. However, among these mothers, the majority (2155) (73.8%) belong to rural residents, while the remaining 767 (26.2%) are urban residents. Regarding the age distribution of mothers, most (1558, 53.3%) were in the age interval of 25–34, followed by 755 (25.8%) in 15–24 and 609 (20.8%) above 34 years. When compared to other categories of factors, the majority of these sampled mothers were illiterate (44.4%), married (93.1%), and the rich (47.5%) (Table 1).

## Log-rank test

A log-rank test was performed to check whether or not significant survival experience differences exist among categories of factors. The null hypothesis states that there is no significant difference between the survival experiences of different groups of categorical variables. Accordingly, there were significant differences in the early initiation of antenatal care visits among the categories of residence, wealth index, age of mothers, and mother's education level (Table 2).

## Testing the proportional hazards assumption

In survival analysis it is important to check whether proportional hazards assumption is satisfied. Since the p-value for the global test was statistically significant at the 5% level of significance, there is statistical evidence to say that the proportional hazards assumption was not satisfied (Table 3).

**Table 2. Log-rank test.**

| Test of Equality over Strata | | | |
|---|---|---|---|
| Variables | Chi-Square | DF | Pr >Chi-Square |
| Residence | 176 | 1 | < .0001 |
| Marital status | 5.7 | 3 | 0.100 |
| Age | 9.9 | 2 | 0.007 |
| Education | 184 | 3 | < .0001 |
| Wealth index | 157 | 4 | < .0001 |

**Table 3. Testing the proportional hazards assumption.**

| The global test | | | |
|---|---|---|---|
| Variables | Chi-Square | DF | Pr >Chi-Square |
| Residence | 13.895 | 1 | 0.00019 |
| Marital status | 3.591 | 3 | 0.30921 |
| Age | 0.197 | 2 | 0.90626 |
| Education | 0.746 | 3 | 0.86237 |
| Wealth index | 12.344 | 4 | 0.01497 |
| Birth order | 3.525 | 1 | 0.06044 |
| Global | 36.32 | 14 | 0.0093 |

**Table 4. AFT model comparison.**

| AFT model comparison | | | |
|---|---|---|---|
| AFT model | AIC | BIC | AICC |
| Exponential | 4875.653 | 4966.727 | 4875.903 |
| Weibull | 4047.225 | 4143.991 | 4047.507 |
| Log-logistic | 3789.606 | 3886.372 | 3789.887 |
| Lognormal | 3795.061 | 3891.827 | 3795.343 |
| Gamma | 3775.749 | 3878.207 | 3776.064 |

## Accelerated failure time models

Since the proportional hazards model was not satisfied, the gamma accelerated failure time model was the best to describe the time to first antenatal care visits among mothers in Ethiopia due to its minimum AIC (3775.749) and BIC (3878.207) values (Table 4).

**The final gamma accelerated failure time model.** Results from the final gamma accelerated failure time model revealed that age of mother, education level, place of residence, wealth index and birth order were the significant predictors of early initiation of antenatal care visits in Ethiopia (Table 5).

## Discussion

The median time to the first antenatal care visit was 4 months, which was above the World Health Organization's definition of early initiation of an antenatal care visit. Moreover, 62% of pregnant women in Ethiopia initiated their antenatal care visit early. This proportion is lower

**Table 5. Results for the gamma AFT model.**

| Variable | Categories | Coefficient | SE | Acceleration factor (φ) | Pr > \|t\| | Lower | Upper |
|---|---|---|---|---|---|---|---|
| Intercept | | 1.3149 | 0.1061 | 3.724 | < .0001 | 3.0250 | 4.5855 |
| Education | No education | 0 | . | 1 | . | . | . |
| | primary | -0.0452 | 0.0318 | 0.956 | 0.1550 | 0.8982 | 1.0172 |
| | secondary | -0.0356 | 0.0513 | 0.965 | 0.4885 | 0.8728 | 1.0672 |
| | Higher | -0.1866 | 0.0622 | 0.830 | 0.0027 | 0.7345 | 0.9373 |
| Age | 15–24 | 0 | . | 1 | . | . | . |
| | 25–34 | 0.526 | 0.0419 | 1.692 | < .0001 | 1.5588 | 1.8369 |
| | >34 | 0.602 | 0.0550 | 1.826 | < .0001 | 1.6392 | 2.0336 |
| Wealth index | Poor | 0 | . | 1 | . | . | . |
| | Middle | -0.0231 | 0.0387 | 0.977 | 0.5514 | 0.9058 | 1.0542 |
| | Rich | -0.1404 | 0.0324 | 0.869 | < .0001 | 0.8156 | 0.9259 |
| Residence | Rural | 0.2143 | 0.0321 | 1.239 | < .0001 | 1.1633 | 1.3195 |
| | Urban | 0 | . | 1 | . | . | . |
| Birth order | Count | 0.7954 | 0.0810 | 2.215 | < .0001 | 1.8901 | 2.5966 |
| PBI | In months | -0.0008 | 0.0005 | 0.999 | 0.0928 | 0.9982 | 1.0001 |
| Marital status | single | -0.2290 | 0.1806 | 0.795 | 0.2049 | 0.5582 | 1.1333 |
| | Married | -0.0565 | 0.0810 | 0.945 | 0.4853 | 0.8064 | 1.1076 |
| | Other | 0 | . | 1 | . | 0 | 0 |
| Scale | | 0.5988 | 0.0128 | | | 1.7757 | 1.8673 |
| Shape | | -0.3731 | 0.0832 | | | 0.5850 | 0.8106 |

Key: PBI, preceding birth interval.

than the prevalence of early initiation of antenatal care visits among low and middle-income countries like Comoros (63.4%), Senegal (64.4%), Ghana (65.7%), Gabon (66.9%), and Liberia (68.6%). This difference in the time to first antenatal care visit may be due to the difference in the age of mother, her education level, wealth index, place of residence, and the birth order of the current pregnancy. The positive value for the regression coefficient (acceletation factor φ>1) implied that the corresponding predictor variable prolonged or accelerated the time to the first antenatal care visit among mothers in Ethiopia.

The estimated acceleration factor for mothers having higher education was φ = 0.830 (p-value = 0.0027), indicating that the time to first antenatal care visit among pregnant women with higher education was 0.830 times shorter than the time to first antenatal care visit among pregnant women with no education. This result is consistent with the previous studies conducted in different parts of the world [23–30]. This may be due to the fact that educated mothers are more likely to understand the advantage of early initiation of antenatal care visits in order to get a better health condition for themselves and their child because they have a greater chance of media and information exposure and a greater decision-making power over their health [31].

The estimated acceleration factors for pregnant women aged 25–34 and above 34 were φ = 1.692 and 1.826, respectively (p-value < .0001), indicating that the time to first antenatal care visit among pregnant women aged 25–34 and above 34 was 1.692 and 1.826, respectively, times longer than the time to first antenatal care visit among women aged 15–24 years. The result is supported by the findings in Ethiopia [23,32]. This may be due to the fact that younger mothers are more concerned about their own and their children's health, more aware of the benefits of early antenatal care visits, and more likely to be persuaded to seek the proper prenatal care.

The estimated acceleration factor for rich women was $\phi$ = 0.869 (p-value < .0001), indicating that the time to first antenatal care visit among rich women was 0.869 times shorter than the time to first antenatal care visits among poor women. The result is consistent with the findings in Ethiopia [13,24,32], Ghana [33], and Cameron [34]. Even though antenatal care visits are free of charge, in order to get this service, there may be direct and indirect costs. Thus, mothers with a higher level of income are more likely to afford these types of costs and start their antenatal care visits earlier than mothers from lower-income families [35].

The estimated acceleration factor for rural women was $\phi$ = 1.239 (p-value < .0001), indicating that the time to first antenatal care visit among rural women was 1.239 times longer than the time to first antenatal visit among urban women. This result is consistent with the studies conducted in Ethiopia [13,23,32], Nigeria [29], and Uganda [30]. This may be due to the greater availability and accessibility of health facilities, health professionals, and health-related information in urban areas than in rural areas.

The estimated acceleration factor for birth order of the current pregnancy was $\phi$ = 2.215 (p-value < .0001), indicating that a one more increment in birth order is associated with a 2.215 times longer time to the first antenatal care visit among pregnant women in Ethiopia. This result is consistent with the findings in Ethiopia [36]. This might be due to the fact that young mothers give more attention and care to their first pregnancy and start their antenatal care visits earlier than mothers with multiple birth histories.

## Conclusion

In Ethiopia, a higher proportion of urban and educated pregnant women (62%) initiated their antenatal care visits early compared to the other groups. Since the proportional hazards assumption was not satisfied, different parametric accelerated failure time models were compared. The gamma-accelerated failure time model was the best fit to the data due to its minimum value of Akaike information criteria and Bayesian information criteria. Moreover, the higher birth order of the current pregnancy, pregnant women being rural residents, women aged 25–34, and women aged above 34 years were associated with an accelerated time to initiation of antenatal care visits. However, pregnant women who attained higher education and rich wealth index were associated with a shorter time to initiation of antenatal care visits. Thus, stakeholders at the federal and regional levels should focus on providing special concern for information, education, and communication regarding the importance of early initiation of antenatal care visits. Moreover they should enforce the provision" education for women".

## Strengths and weaknesses of the study

This study demonstrates how to use the data collected through demographic and health surveys. The methodology used to identify factors influencing initiation of antenatal care visits is correct and the results are in line with known results. However, the data must be integrated with the other fixed and time varying covariates, which can be collected by ad hoc surveys on clinical records, to examine the changes over time within individuals and to justify a delayed initiation of antenatal care visits. Although, it is possible to conduct multilevel analysis using the DHS data, this study is a single level survival analysis.

## Acknowledgments

The authors would like to thank the Central Statistical Agency of Ethiopia for making the data freely available for research purposes. Moreover, we are very grateful to the institutional review board of the International Classification of Functioning (ICF) for their support in providing an ethical statement.

## Author Contributions

**Conceptualization:** Nuru Mohammed Hussen, Kindu Kebede Gebre, Tilahun Gemechu Rorisa, Mekash Ayalew Mohammed.

**Methodology:** Nuru Mohammed Hussen.

**Project administration:** Nuru Mohammed Hussen.

**Resources:** Nuru Mohammed Hussen.

**Software:** Nuru Mohammed Hussen.

**Supervision:** Nuru Mohammed Hussen.

**Validation:** Nuru Mohammed Hussen.

**Visualization:** Nuru Mohammed Hussen.

**Writing – original draft:** Nuru Mohammed Hussen.

**Writing – review & editing:** Nuru Mohammed Hussen.

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
