## [Decision Letter · Decision Letter 0]

21 Jun 2024

PONE-D-24-01074Multilevel analysis of early initiation of antenatal care visits and associated factors in EthiopiaPLOS ONE

Dear Dr.  Hussen,

Thank you for submitting your manuscript to PLOS ONE. After careful consideration, we feel that it has merit but does not fully meet PLOS ONE’s publication criteria as it currently stands. Therefore, we invite you to submit a revised version of the manuscript that addresses the points raised during the review process.

We look forward to receiving your revised manuscript.

Kind regards,

Agegnehu Bante

Academic Editor

PLOS ONE

Journal Requirements:

Reviewers' comments:

Reviewer's Responses to Questions

**Comments to the Author**

1. Is the manuscript technically sound, and do the data support the conclusions?

Reviewer #1: Yes

Reviewer #2: No

Reviewer #3: Partly

2. Has the statistical analysis been performed appropriately and rigorously? 

Reviewer #1: Yes

Reviewer #2: No

Reviewer #3: Yes

3. Have the authors made all data underlying the findings in their manuscript fully available?

Reviewer #1: Yes

Reviewer #2: Yes

Reviewer #3: Yes

4. Is the manuscript presented in an intelligible fashion and written in standard English?

Reviewer #1: Yes

Reviewer #2: Yes

Reviewer #3: No

5. Review Comments to the Author

Reviewer #1: I appreciate this work for its novel concept and well-written content, which can offer new insights to readers. However, I have some comments and questions in the following section.

INTRODUCTION

“Moreover, an evidence-based report of the World Health Organization (WHO) recommended eight antenatal visits among pregnant women in low- and middle-income countries, where the first visit should be before week 16, the second visit should be between weeks 24 and 28, the third visit should be between weeks 30 and 32, and the fourth visit should be between 36 and 38 weeks of pregnancy” This sentence in the introduction part is not clear, you mentioned WHO recommends 8 contacts, but the contact times seems similar to those of focused antenatal care visiting times or do you mean that WHO recommends only four visiting times for LMICs?. Additionally, it would be better to use consistent terminology, as the word "visit" has been replaced by "contact.

Sampling design

There is a discrepancy between your report and the 2019 Mini EDHS regarding the number of interviewed women of reproductive age. Your paper states, "The survey interviewed 5,753 women of reproductive age (15–49) from a nationally representative sample of 8,663 households," while the 2019 Mini EDHS indicates that out of 9,012 women, 8,885 were eligible and interviewed. How did these inconsistencies arise? Could you please provide your evidence?

Thank you

Reviewer #2: Title

1. Who are your study participants? Among whom is your study conducted? It is mandatory to highlight your participants in your study title.

Abstract

Method section: It is better to include the study design. What is your study’s design? Is it a cross-sectional study or retrospective cohort study?

Result:

1. Do you think the word prevalence is appropriate for your study?

2. Could you explain me why the point estimate and interval estimate were different? The point estimate shows positive relation whereas, the period estimate shows negative association.

3. I think it is better and appropriate to change the word “lower time to initiate” by “shorter time to initiate”.

Keyword:

1. You need to incorporate the study population.

Introduction

1. Where did you find this WHO recommendation for the early initiation of ANC visits in low and middle income countries with in the first 16 weeks? The reference you used for this statement is not right. The reference deals about intrapartum care. Where did you get this recommendation? Moreover, is it a recent recommendation? Otherwise, you should rely on the 12 weeks recommendation forwarded by WHO in 2016 for positive pregnancy experience.

2. Needs major revision. E.g. the background section is not focused to the issue under study.

Methods

1. The study design used for the EMDHS was a cross-sectional study. So, what is the study design you employed for your study? You need to mention your study design.

2. Explanatory variables for the outcome variable should be exhaustively listed. You need to cite from where you got those explanatory variables.

3. Your explanatory are too few. I am in doubt how you fitted your model?

4. Data processing and analysis should be discussed as a subsection of the method part separately from study variables (explanatory variables). In addition, you need to present this subsection in much details.

5. Where are the community level variables you considered for your study? How can it be a multi-level study without including community-level factors? This a big issue that you need to respond.

6. What is censoring?

7. Gamma accelerated failure time model why? A known survival analysis can estimate the time to first ANC.

8. Have you used weighting? What type of weighting? How the weighting variable is created? If there is no weighting – it is a must to have weighted analysis because the sampling strategy of DHS necessitates the use of weighting.

Result

1. What new findings you have reported?

2. Where is the model comparison section? Have you compared and ascertained that the model with both individual and community level factors is the best fit model? I don’t think so. This is a major problem.

Discussion:

1. Strength and limitations: Should exactly describe the limitation and strengths of the paper. You need to acknowledge your limitations.

Reviewer #3: The objective of this study was to identify the factors associated with early initiation of antenatal care (ANC) visits in Ethiopia, which is important for early detection of high-risk pregnancies and complications. Data was collected from 2922 mothers in a population-based cross-sectional survey, and a gamma-accelerated failure time model was used to analyze factors associated with early initiation of ANC, which was defined as the number of weeks to the first ANC visit following conception. The study found that higher birth order of the current pregnancy, rural residence, and mothers aged 25-34 and above 34 were associated with a longer time to initiation of ANC visits, while higher education and wealth index were associated with earlier initiation. The authors recommend that stakeholders focus on providing more information, education, and communication regarding the importance of early initiation of ANC visits in Ethiopia.

There are some basic adjustments required so you can take serious steps:

1- There is no need to create a separate section for Operational definitions that can be explained along with the details of the methodology.

2- There is no need to explain the different distributions. It is better to focus on the applied model in the analysis and justify the reason for its choice. It is not clear why the gamma-accelerated failure time model is preferred over others, and where are the references that confirm this.

3- Where is the detailed explanation of the Gamma-accelerated failure time model and the meaning and interpretation of the parameters?

4- Researchers must write the steps of the analysis, the names of the tests that will be applied, the reason for applying them, and the subsequent steps that follow. The researchers have already done this in the results section, but the steps must be written in the methodology section before application.

5- Tables are poorly organized and need to be modified

6- Why did the researchers not combine divorced and widowed persons into one category under the marital status variable, since the percentage of cases is less than 5%?

7- The discussion needs to be developed in more depth

8- The conclusion is poorly written. The researchers talked about the prevalence of receiving primary care among educated and urban women, and suddenly they talked about the hypotheses of the model. I hope that researchers review the writing style, the flow of sentences, and the logical order in writing.

9- Some limitations of this study could include the use of secondary data, which may not capture all relevant factors for analysis, and the fact that the study did not include important predictors like previous birth histories due to data availability.

10- The strengths and weaknesses of the paper do not include all elements, and need to be developed and modified

6. PLOS authors have the option to publish the peer review history of their article (what does this mean?). If published, this will include your full peer review and any attached files.

Reviewer #1: No

Reviewer #2: No

Reviewer #3: **Yes: **Suzan Abdel-Rahman

---

## [Author Response · Author response to Decision Letter 0]

1 Jul 2024

Title: survival analysis of early initiation of antenatal care visits and associated factors among pregnant Women in Ethiopia

Response to reviewers

Reviewer #1

Thanks for spending your golden time and providing constructive comments on this manuscript.

Question 1: 

Introduction:

“Moreover, an evidence-based report of the World Health Organization (WHO) recommended eight antenatal care visits among pregnant women in low-and middle-income countries, where the first visit should be before week 16, the second should be between week 24 and 28, the third visit should be between week 30 and 32, and the fourth visit should be between 36 and 38 weeks of pregnancy”. This sentence in the introduction part is not clear, you mentioned WHO recommended 8 contacts, but the contact time seems to those focused antenatal care visiting times for LMICs? Additionally, it would be better to use consistent terminology, as the word “visit” has been replaced by”contact”.

Answer 1: First I would like to thank for your interesting and constructive question. I have taken this question as a comment and incorporate in the revised version of the manuscript. The statement is related to the previous statement. Where the first statement indicates the recommendation of at least four antenatal care visits among pregnant women. But the underlying evidence is concerned with the recommendation of eight antenatal care visits rather than four antenatal care visits. The paragraph is corrected as follows;

“The WHO recommended four or more visits among pregnant women in low- and middle-income countries, where the first visit should be between week 8 and 12, the second visit should be between weeks 24 and 26, the third visit should be at week 32, and the fourth visit should be between 36 and 38 weeks of pregnancy and return for delivery at week 41 if the women doesn’t give birth. Moreover, an evidence-based report of the World Health Organization (WHO) recommended eight rather than four antenatal care visits among pregnant women in low- and middle-income countries.” Finally, there were conditions where both words “visits” and “contact” were interchangeably used, but for the issue of consistency the word “contact” was replaced by “visit” in the revised manuscript. 

Question 2: 

Sampling design:

There is a discrepancy between your report and the 2019 Mini EDHS regarding the number of interviewed women of reproductive age. Your paper states” the survey interviewed 5753 women of reproductive age (15-49) from a nationally representative sample of 8663 households,” while the 2019 Mini EDHS indicates that out of 9012 women,8885 were eligible and interviewed. How did these inconsistencies arise? Could you please provide your evidence?

Answer 2: I have no word to thank for your question regarded as a comment and which help me to revise the report and the datasets again. The data set and the report realizes what you have said, then after incorporating your comment this statement is corrected as:

“8,885 women of reproductive age (15–49) were eligible and interviewed from a nationally representative sample of 9012 women.” 

Reviewer #2

Thanks for spending your golden time and providing constructive comments on this manuscript.

Question 1:

Who are your study participants? Among whom is your study conducted? 

It is mandatory to highlight your participants in your study tittle.

Answer 1: thanks for your interesting and important questions and comment. Our study participants were the weighted sample of 2922 pregnant women with complete information about their first time of antenatal care visits. Even though, these women gives a complete information about their first time to antenatal care visit, 8885 women were eligible and interviewed during the survey. By incorporating this comment the tittle was modified as:

“Survival analysis of early initiation of antenatal care visits and associated factors among pregnant women in Ethiopia”

Question 2: 

Abstract

Methods part: it is better to include the study design. What is your study’s design? Is it a cross-sectional study or retrospective cohort study?

Answer 2: thanks for your question. It is clear that this study is a retrospective cohort study, because it uses the data collected in 2019, but the 2019 mini EDHS was population based cross-sectional survey, collected in a specific point in time to solve a specific problems. For this matter the study design was included in the revised manuscript.

Question 3:

Result part: do you think the word prevalence is appropriate for your study.

Answer 3: In many studies both the percentages of early initiation and delayed initiation of antenatal care visits are presented as a prevalence. But after this question the term prevalence is more appropriate for delayed initiation of ANC visit rather than early initiation. Then we try to incorporate this comment in the revised manuscript.

Question 4: Could you explain me why the point estimate and interval estimate different? The point estimate shows positive relation, whereas the period estimate shows negative association.

Answer 4: Thanks for your relevant and professional question. We have clearly observe the difference in conclusion among point and interval estimates, but this is not correct. This error occurs, because the confidence intervals were for the coefficients, but in the revised version of our manuscript we present the point estimates with their confidence interval estimates for the acceleration factor.

Comment1: 

Keyword: you need to incorporate the study population.

Response1: We have incorporated this comment in the revised version of our manuscript.

Question5: 

Introduction:

1. Where did you find this WHO recommendation for early initiation of ANC visits in low and middle income countries with in the first 16 weeks? The reference you used for this statement is not right. The reference you deals about intrapartum care. Where did you get this recommendation? Moreover, is it recent recommendation? Otherwise, you should rely on the 12 weeks recommendation forwarded by WHO in 2016 for positive pregnancy experience.

Answer 5: 

Thanks for this important question, the evidence cited in the manuscript regarding initiation of antenatal care visits in LMICs have no sufficient evidence including the one cited in the paper. Then by considering this truth we have cited the most recent evidence in the revised version of the manuscript.

Methods:

Question 6: The study design used for the EMDHS was cross-sectional study. So what is the study design you employed for your study? You need to mention your study design.

Answer 6: the study presented the study design of the EMDHS as a cross-sectional study design. But the study design employed for this study is a retrospective cohort study, because it uses the past data in 2019 and since the study included the mothers with a complete information about their first antenatal care visit.

Question 7: 

Explanatory variables for the outcome variables should be exhaustively listed. You need to cite from where you got those explanatory variables?

 Answer 7: you are right, the selection of explanatory variables to a given response variable should be based on the existing literatures. The predictor variables included in this study were significantly associated with initiation of antenatal care visits according to different studies listed in the introduction part of this study.

Comment 3:

Data processing and analysis should be discussed as a sub section of the method part separately from the study variables (Explanatory variables).

Response 3: We found this comment constructive and incorporated in the revised version of the manuscript.

Question 8:

Where are the community level variables you considered for your study? How can it be a multilevel analysis without including community level factors? This is a big issue that you need to respond.

Answer 8: thanks for this important question. Even though the tittle of our study said multilevel analysis, the steps in the paper organization is single level survival analysis. Then we have modified the tittle, but it is possible to conduct two-level and three-level analysis using the underlying data.

Question 9:

What is censoring?

Answer 9: 

Before we define what censoring is, let’s define what survival analysis is?

Survival analysis is a collection of statistical procedures for data analysis for which the outcome variable of interest is time until an event occurs. However. Censoring is the basic term in survival analysis which censoring occurs when we have some information about individual survival time, but we don’t know the survival time exactly. In most cases censoring occurs due to the following reasons;

1. a person does not experience the event before the study ends;

2. a person is lost to follow-up during the study period;

3. a person withdraws from the study because of death some other reason 

When we come to our study, it concerned with the time to first antenatal care visit and according to WHO recommendation early initiation of ANC visit is in the first 16 weeks of pregnancy. Then when women starts their ANC in this period it is considered as event and if it is beyond WHO recommendation it is considered as Censoring.

Question 10: gamma accelerated failure time model why? A known survival analysis can estimate the time to first ANC.

Answer 10:

Yes we may get a result about initiation of ANC using the standard survival analysis, but there are two conditions to fit the model;

1. We select the best model that fits the data well and with a minimum error.

2. We should check for a certain model assumptions and if violated take a recommended action. 

In our study, firstly we have fitted the Cox PH model and check for the assumption the hazard ratios are constant over the survival times using the global test. Unfortunately, this assumption was not satisfied, then the next step is to fit the accelerated failure time model with different distributions for the survival time and select the one with the best fit. The gamma AFT model fits the data well.

Question 11:

Have you used weighting? What type of weighting? How the weighting variable is created? If there is no weighting- it is a must to have weighted analysis because the sampling strategy of DHS necessitates the use of weighting.

Answer 11:

As stated in the methodology part of the manuscript, EDHS data was collected from all regions in the country from the selected enumeration areas. But the numbers of enumeration areas selected from each region are not equal, the number of households selected from each enumeration area are not equal. Then weighting is necessary to perfectly reflect the demographic characteristics of the entire population. Weighting helps to minimize the bias, by adjusting for any imbalances in the sample and produce estimates that are more representative of the population as a whole. Then we have used sampling weights which are assigned to each women to account for the probability of selection in the sample. These weights are calculated as the inverse of the probability of selection taking into account stratification, clustering and unequal selection probabilities. As stated in the manuscript weighting was made using SPSS. We have created weight variable by dividing the variable “Women’s individual sample weight (V005)” by 1,000,000 then we weight the data using this variable.

Question 12:

What new findings you have reported?

Answer 12:

Our objective here is to identify the key factors associated with early initiation of antenatal care visits using survival analysis, then our finding should identify the factors associated with early initiation of ANC visits. Then the factors associated with early initiation of ANC visits were, age of mothers, birth order of the current pregnancy, mother’s education level, wealth index, and place of residence.

Comment 4:

Strength and limitations: should exactly describe the limitation and strengths of the paper. You need to acknowledge your limitations.

Response 4: 

Thanks for this constructive comment, we have incorporated this comment while we provide the revised manuscript.

 Reviewer #3

Thanks for spending your golden time and providing constructive comments on this manuscript.

Comment 1: 

There is no need to create a separate section for operational definitions that can be explained along with the details of the methodology.

Response 1: 

Thanks for your important comment, we have incorporated this comment while preparing the revised manuscript.

Comment 2:

There is no need to explain the different distributions. It is better to focus on the applied model in the analysis and justify the reason for this choice. It is not clear why the gamma accelerated failure time model is preferred over others, and where is the reference that confirms this.

Response 2:

In the methodology section of our manuscript, we have proposed different distributions for the survival times if the PH assumption is not satisfied. Then in the result section it is time to identify which distribution fits better than the remaining distributions. The in our model comparison section since the models with different distributions was non nested model, the best model is the one with the minimum value of the fit statistic values (AIC, BIC, AICC, BICC). Then among these models the accelerated failure time model with the gamma distribution has the minimum values for the fit statistics, then it is fitted in the final analysis.

 Question 1:

Where the detailed explanation of the gamma is accelerated failure time model and the meaning and interpretation of parameters?

Answer 1:

We have included this section in the methodology section of the revised manuscript. 

Comment 3:

Researchers must write the steps of the analysis, the names of the tests that will be applied, the reason for applying them, and the subsequent steps that follow. The researchers have already done this in the results section, but the steps must be written in the methodology section before application.

Response 3:

We have incorporated this important comment while preparing the revised manuscript.

Question 2:

Why did the researchers not combine divorced and widowed persons into one category under the marital status variable, since the percentage of cases is less than 5%?

Answer 2:

Thanks for your professional question, the authors aware of merging categories with low frequencies, used to increase the sample size within each merged categories, which improves the precision of estimates and increase the likelihood of detecting meaningful associations in the data. Having this awareness we have combined these categories in the revised version of our manuscript. 

Comment 4:

Tables are poorly organized and needs to be modified.

Response 4:

We have taken actions in the revised manuscript 

Comment 5:

The discussion needs to be developed in more depth.

Response 5:

We try to incorporate the comment, in the revised manuscript.

Comment 6:

The conclusion is poorly written. The researchers talked about the prevalence of receiving primary care among educated and urban women, and suddenly they talked the hypothesis of the model. I hope that the researchers review the writing style, the flow of the sentences, and the logical order in writing.

Response 6:

We have incorporated this comment in the revised manuscript.

Comment 7:

The strengths and weaknesses of the paper do not include all elements, and need to be developed and modified.

Response 7:

We try to incorporate this comment in the revised manuscript.

---

## [Decision Letter · Decision Letter 1]

24 Oct 2024

PONE-D-24-01074R1Survival analysis of early initiation of antenatal care visits and associated factors among pregnant women in EthiopiaPLOS ONE

Dear Dr. Hussen,

Thank you for submitting your manuscript to PLOS ONE. After careful consideration, we feel that it has merit but does not fully meet PLOS ONE’s publication criteria as it currently stands. Therefore, we invite you to submit a revised version of the manuscript that addresses the points raised during the review process.

We look forward to receiving your revised manuscript.

Kind regards,

Agegnehu Bante

Academic Editor

PLOS ONE

Journal Requirements:

Reviewers' comments:

Reviewer's Responses to Questions

**Comments to the Author**

1. If the authors have adequately addressed your comments raised in a previous round of review and you feel that this manuscript is now acceptable for publication, you may indicate that here to bypass the “Comments to the Author” section, enter your conflict of interest statement in the “Confidential to Editor” section, and submit your "Accept" recommendation.

Reviewer #3: (No Response)

Reviewer #4: All comments have been addressed

Reviewer #5: All comments have been addressed

2. Is the manuscript technically sound, and do the data support the conclusions?

Reviewer #3: Yes

Reviewer #4: Yes

Reviewer #5: Yes

3. Has the statistical analysis been performed appropriately and rigorously? 

Reviewer #3: Yes

Reviewer #4: Yes

Reviewer #5: Yes

4. Have the authors made all data underlying the findings in their manuscript fully available?

Reviewer #3: Yes

Reviewer #4: No

Reviewer #5: Yes

5. Is the manuscript presented in an intelligible fashion and written in standard English?

Reviewer #3: Yes

Reviewer #4: Yes

Reviewer #5: Yes

6. Review Comments to the Author

Reviewer #3: Researchers need to understand the distinction between the methodology section and the results section. All explanations of statistical methods should be included in the methodology section, while the results section should only present the results. I previously requested this but it was not understood.

Reviewer #4: The authors have successfully addressed all the concerns raised by the reviewers, so I recommend its publication in PLOS ONE.

Reviewer #5: The authors have revised the manuscript properly by incorporating all of the corrections/suggestion, therefore, it is recommended for publication in its present form.

7. PLOS authors have the option to publish the peer review history of their article (what does this mean?). If published, this will include your full peer review and any attached files.

Reviewer #3: **Yes: **Suzan Abdel-Rahman

Reviewer #4: No

Reviewer #5: No

---

## [Author Response · Author response to Decision Letter 1]

28 Oct 2024

Response to reviewers and editors

A. Response to Editor

Comment:

Response:

Thanks for this constructive comment, which help us to revise our reference list. Hence we have assured that no retracted papers have been cited in our reference list.

B. Response to reviewers

Reviewer #3

Thanks again for spending your golden time and providing constructive comments on this manuscript.

Comment 1: 

Researchers need to understand the distinction between the methodology section and the results section. All explanations of statistical methods should be included in the methodology section, while the results section should only present the results. I previously requested this but it was not understood.

Response 1:

We have incorporated this constructive comment while preparing the revised version of this manuscript.

---

## [Decision Letter · Decision Letter 2]

1 Dec 2024

Survival analysis of early initiation of antenatal care visits and associated factors among pregnant women in Ethiopia

PONE-D-24-01074R2

Dear Dr. Hussen,

We’re pleased to inform you that your manuscript has been judged scientifically suitable for publication and will be formally accepted for publication once it meets all outstanding technical requirements.

Kind regards,

Agegnehu Bante

Academic Editor

PLOS ONE

Additional Editor Comments (optional):

Reviewers' comments:

Reviewer's Responses to Questions

**Comments to the Author**

1. If the authors have adequately addressed your comments raised in a previous round of review and you feel that this manuscript is now acceptable for publication, you may indicate that here to bypass the “Comments to the Author” section, enter your conflict of interest statement in the “Confidential to Editor” section, and submit your "Accept" recommendation.

Reviewer #5: All comments have been addressed

2. Is the manuscript technically sound, and do the data support the conclusions?

Reviewer #5: Yes

3. Has the statistical analysis been performed appropriately and rigorously? 

Reviewer #5: Yes

4. Have the authors made all data underlying the findings in their manuscript fully available?

Reviewer #5: Yes

5. Is the manuscript presented in an intelligible fashion and written in standard English?

Reviewer #5: Yes

6. Review Comments to the Author

Reviewer #5: The authors have incorporated all of the suggestions and corrections, therefore, it is accepted in its present form.

7. PLOS authors have the option to publish the peer review history of their article (what does this mean?). If published, this will include your full peer review and any attached files.

Reviewer #5: No

---

## [Editor Report · Acceptance letter]

18 Dec 2024

PONE-D-24-01074R2 

PLOS ONE

Dear Dr. Hussen, 

I'm pleased to inform you that your manuscript has been deemed suitable for publication in PLOS ONE. Congratulations! Your manuscript is now being handed over to our production team.

Kind regards, 

on behalf of

Mr. Agegnehu Bante 

Academic Editor

PLOS ONE